# Metallic and Non-Metallic Plasmonic Nanostructures for LSPR Sensors

**DOI:** 10.3390/mi14071393

**Published:** 2023-07-08

**Authors:** Judy Z. Wu, Samar Ali Ghopry, Bo Liu, Andrew Shultz

**Affiliations:** 1Department of Physics and Astronomy, University of Kansas, Lawrence, KS 66045, USA; jwu@ku.edu (J.Z.W.); andjshultz@ku.edu (A.S.); 2Department of Physics, Jazan University, Jazan 45142, Saudi Arabia

**Keywords:** graphene, 2D materials, plasmonic nanostructures, LSPR, SERS, high sensitivity

## Abstract

Localized surface plasmonic resonance (LSPR) provides a unique scheme for light management and has been demonstrated across a large variety of metallic nanostructures. More recently, non-metallic nanostructures of two-dimensional atomic materials and heterostructures have emerged as a promising, low-cost alternative in order to generate strong LSPR. In this paper, a review of the recent progress made on non-metallic LSPR nanostructures will be provided in comparison with their metallic counterparts. A few applications in optoelectronics and sensors will be highlighted. In addition, the remaining challenges and future perspectives will be discussed.

## 1. Introduction

The discovery of graphene by Andre Geim and Konstantin Novoselov in 2004 (Nobel Prize in Physics in 2010), and many other two-dimensional atomic materials (2D materials) afterwards, provided a new platform for the design of sensors with high levels of sensitivity while at a low cost. The strong quantum confinement in these 2D atomic materials renders unique electronic structures and physical properties possible that are not available in conventional semiconductor bulks and films. For example, graphene—a monolayer of carbon atoms arranged in a flat 2D honeycomb lattice [1,2]—has attracted enormous interest due to its superior physical properties including high electron and hole mobility, broadband absorption from near UV to middle-infrared, flexibility and chemical stability [1,2]. Furthermore, van der Waals (vdW) heterostructures of the 2D atomic materials provide a platform for further tuning of electronic structures via interface interactions.

Absorption of light is used in a wide range of sensor applications, implementation of nanostructures for the generation of localized surface plasmonic resonance (LSPR) becomes an important topic in research and development of the next generation of sensors. For example, photodetectors with LSPR nanostructures can have an increased light absorption enabled by the light trapping of the LSPR nanostructures and hence enhanced light–solid interaction, which in turn leads to major improvements in photoresponsivity and detectivity. Another example is in biological/chemical sensing using surface-enhanced Raman spectroscopy (SERS). The implementation of the LSPR nanostructures to SERS substrates promotes the so-called electromagnetic enhancement (EM), which can lead to a higher SERS sensitivity by 8–10 orders of magnitude and is suitable for detection of single or a few molecules. The advantages of the LSPR effect have prompted intensive research to explore various nanostructures for sustained plasmonic resonance. Among them, the majority relies on metallic nanostructures; the large charge concentrations in metals leads to plasmonic resonance frequencies, typically in the visible spectrum. The tunability of the electronic structure and charge carrier concentration in 2D materials and their vdW heterostructures make them promising for generating the LSPR effect in a broader spectrum at lower costs. This has motivated more recent research of 2D material-based nanostructures as well as vdW heterostructures, and interesting results have been obtained. It should be noted that these non-metallic LSPR nanostructures have distinctive advantages over their metallic counterparts including high performance, low cost and scalability [3,4,5,6,7]. This review highlights some of the recent progress made in research and development of LSPR nanostructures based on the non-metallic 2D materials and their vdW heterostructures in comparison with the metallic ones. Applications of these LSPR nanostructures in photodetection and SERS will be used as illustrations.

## 2. Basic Physics of LSPR Effect in Metallic and Non-Metallic Nanostructures

LSPR regards confinement of surface plasmons induced by electron oscillations on the surface of nanostructures typically of metallic materials with a large free electron concentration [8,9]. Advancements in the generation of various nanostructures in the last two decades, either using top-down approaches of advanced lithography or bottom-up ones of direct synthesis, has prompted tremendous progress in research and development of LSPR for various applications including photovoltaics, photodetection, sensing, etc. Key in the LSPR effect is the light-induced electron resonance or surface plasmon on the surface of nanoparticles (NPs) (see Figure 1) of dimension comparable or smaller than the wavelength of the incident light. The electron oscillation frequency is determined by the density of electrons, the effective electron mass as well as the size and shape of the charge distribution [10]. This means the resonance frequency can be tuned by selecting different materials and changing the morphology of the nanostructures. Near the surface of LSPR nanostructures, the electric field is significantly enhanced, which falls off quickly with increasing distance from the nanostructure. This means the LSPR is a near-field effect, which requires coupling with the plasmon at a relatively short distance to be effective [11,12]. Another distinctive feature of the LSPR nanostructures is the enhanced light absorption, which reaches maximum at the plasmon resonance frequency, exhibited as a dip in the optical transmission spectrum, which could be used to design LSPR nanostructures for optimal light–solid interaction for sensors that rely on the light absorption.

Theoretically, LSPR is not necessarily limited to metallic nanostructures. This means that semiconductors may also have LSPR effect if adequate carrier concentration could be generated. Figure 2 compares the plasmon frequencies in metals and semiconductors based on the carrier concentrations [13]. A unique advantage of semiconductors is the broad range of the free carrier density of 10^17^–10^22^/cm^3^, which can lead to a broad range of the LSPR frequency from THz to near-infrared (Figure 2). An excellent example is the plasmonic effect in graphene that has low 2D carrier concentration around 10^12^/cm^2^ and hence plasmonic frequency in the middle-infrared (or sub-THz) spectra as illustrated experimentally [14,15]. Even in the visible spectrum, the semiconductor LSPR nanostructures may also present advantages over their metallic counterparts including possible lower Ohmic loss and cost. In fact, the metallic LSPR nanostructures are primarily limited to a few noble metals such as Au, since other metals may form a degraded surface layer in ambient or device operating environments, which is detrimental to LSPR as electron resonance occurs exactly on the surface of the nanostructures.

## 3. Application of LSPR Nanostructures for Surface-Enhanced Raman Spectroscopy (SERS)

### 3.1. Design of SERS Substrates

Raman spectroscopy is a spectroscopic method that employs light to detect different molecules using inelastic scattering of the photons from vibrational modes of the molecules. Physically, there are two possible explanations for Raman scattering: quantum interaction and classical interaction. In the former, by shining light on the molecules, the electrons become excited and change to a virtual energy state before relaxing back to the initial state. The molecule will re-emit the photon with an energy that is either higher (anti-Stokes shifted) or lower (Stokes shifted) than the incident photon. However, only one out of 10^7^ of the excited molecules may land on a different energy level from their original state. This inelastic scattering was discovered by an Indian physicist Raman in 1928 and was named as Raman scattering, in contrast elastic scattering is called Rayleigh scattering. The classical interaction can be understood using a simple diatomic molecule that may represent molecular vibrations and by Hooke’s law to express the displacement of the molecule as in Equation (1):(1)$$\frac{m_{1}m_{2}}{m_{1}+m_{2}}\left(\frac{d^{2}x_{1}}{{dt}^{2}}+\frac{d^{2}x_{2}}{{dt}^{2}}\right)=-k\left(x_{1}+x_{2}\right)$$
where the vibration frequency depends on the spring constant (*k*), and the atomic mass $m_{1}$ and $m_{2}$. Lastly, *x* and k represent the displacement and bond strength, respectively. Using $\mu =\frac{m_{1}m_{2}}{m_{1}+m_{2}}$, and $q=x_{1}+x_{2}$, Equation (2) could be simplified as:(2)$$\mu \left(\frac{d^{2}q}{{dt}^{2}}\right)=-k\left(q\right).$$

The solution to Equation (3) could be given by:(3)$$q=q_{0\;}\cos(2\pi v_{m}t),$$
where $v_{m=}\frac{1}{2\pi }\sqrt{\frac{K}{\mu }}$.

Due to the low cross-section of Raman scattering, which is often more than five orders of magnitude less than Raleigh scattering, the probability of a Raman scattered photon occurring is quite low. As a result, Raman spectroscopy has a fundamental limitation in its sensitivity. When compared to fluorescence, which has a cross-section of about 10^−19^ cm^2^, Raman scattering has a cross-section of about 10^−29^ cm^2^, which is extremely small [16]. This means that Raman spectroscopy would only be able to characterize bulk compounds unless an efficient approach could be found to increase its sensitivity by orders of magnitude.

Surface-enhanced Raman spectroscopy (SERS) is a recently developed method for increasing the cross-section of Raman scattering by orders of magnitude to reach at least 10^−16^ cm^2^ or higher [16,17,18,19,20] and has shown a detection ability of a single molecule [17,21,22]. SERS relies on two major enhancement mechanisms: electromagnetic mechanism (EM) and chemical mechanism (CM). The EM with an enhancement factor up to 10^8^ includes the enhanced local electromagnetic field that generates near the probe molecules via plasmonic nanostructures on SERS substrates [23,24,25]. As shown schematically in Figure 3a using metal NPs as an example [26,27,28], LSPR is the coherent oscillation of free charge carriers induced by illumination at the surface of metal NPs. Molecules positioned close to plasmonic nanostructures are exposed to highly enhanced electromagnetic fields produced by the oscillating electrons on the surface of the nanostructures at distances of a few nanometers to hundreds of nanometers. Quantitatively, the EM enhancement factor may be affected by the plasmonic nanostructure materials, dimension, morphology and carrier density. Furthermore, the dielectric properties of the surrounding medium also play an important role in the optimization of plasmonic nanostructures [29]. EM and CM effects can be both integrated on one SERS substrate by selecting an appropriate substrate such as graphene (Figure 3a).

The CM with enhancement factor of about 10^3^ depends on charge transfer at the interface between the molecule and the SERS substrate [30,31,32]. Quantitatively, the CM enhancement factor is affected by the alignment of the interface electronic band edge that assists the charge transfer between probe molecules and the SERS substrate. To obtain a high CM enhancement factor, SERS substrate must be selected or tunable to enable favorable band-edge alignment based on the probe molecule’s electronic structure with consideration for the positions of the highest-occupied molecular orbital (HOMO) and the lowest-unoccupied molecular orbital (LUMO) at the molecule/SERS substrate interface to allow a significant CM enhancement [33]. Two-dimensional atomic materials offer a promising solution to this need, as they can be transferred or grown on regular SERS substrates [34]. This has led to intensive research on 2D material SERS substrates in the past decade or so. For instance, graphene with Fermi energy at ~4.5 eV is compatible with the HOMO and/or LUMO of a great number of molecules as illustrated schematically in Figure 3b [30,32,35,36]. Another well-studied group of 2D materials is transition metal dichalcogenides (TMDC) [37,38] such as MoS_2_ and WS_2_, which exhibit a transition from indirect-bandgap to direct-bandgap semiconductor when approaching monolayer from bulk and an enhanced light–solid interaction as needed for LSPR.

**Figure 3 micromachines-14-01393-f003:**
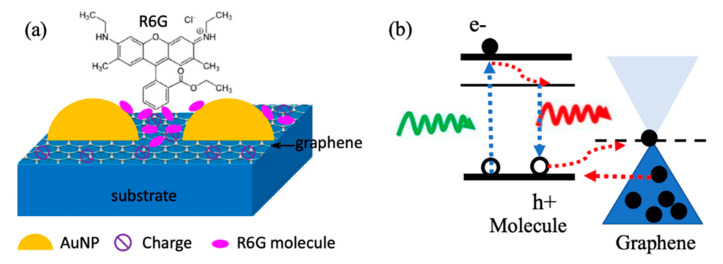
(**a**) Schematic description of R6G molecules attached on Au nanoparticles/graphene SERS substrate. Reproduced with permission [39]. Copyright 2015, Elsevier. (**b**) Schematic description of the SERS process with graphene-enhanced CM effect.

### 3.2. SERS Substrates Based on Non-Metallic 2D vdW Heterostructures

It should be noted that the majority of the reported LSPR nanostructures adopt metallic materials like Au and Ag. Since Ag nanostructures typically degrade in ambient conditions over a period of 1–2 weeks, Au is actually the material most frequently employed in SERS substrates. On top of being expensive, Au LSPR nanostructures may also be lossy, which means that the achieved SERS EM enhancement is a compromise between the ohmic loss on Au nanostructures and the LSPR. Therefore, the examination of non-metallic nanostructures for SERS is important.

Two-dimensional atomic materials, such as graphene and TMDC, can be stacked together to create vdW heterostructures [40]. This allows tuning of the electronic structures of these 2D vdW heterostructures beyond the limit of conventional bulks. Specifically for SERS, these 2D vdW heterostructures may provide a unique platform for SERS substrates with possible SERS EM enhancement which is facilitated by the dipole−dipole interaction at the interface between the layers of vdW heterostructures. A high LSPR effect may be obtained on non-metallic 2D vdW heterostructure via photodoping enabled by the TMDC/graphene interfacial dipole−dipole interaction, resulting in a surprising ultra-high resonant SERS sensitivity. In recent work, Ghopry et al. reported strong EM enhancement attributed to the LSPR effect on TMDC nanodisc (TMDC-ND)/graphene vdW heterostructures [41]. It should be noted that single-layer-transferred graphene (or directly created graphene [42]) on SiO_2_/Si wafers can be coated layer by layer with TMDC nanostructures to create the TMDC nanostructures/graphene vdW heterostructures utilizing a chemical vapor deposition (CVD) technique [41,43,44]. Additionally, by controlling the growth conditions, as shown schematically in Figure 4a–g, diverse morphologies, including nanodisc (N-disc) [41,44] and nanodonuts (N-donut) [43], can be produced.

Figure 5a displays the Raman map of MoS_2_ nanostructure using A_1g_ mode [43]. The AFM images of the sample confirm the morphology and concentration (Figure 5b,c) as well as the size of the MoS_2_ N-disc and N-donut. The high SERS sensitivity of R6G probe molecules (Figure 5e,f) in this study demonstrates that 2D TMDC nanostructure/graphene vdW heterostructures may offer a distinctive platform for designing new SERS substrates with both EM and CM enhancements [33,45,46,47]. Density functional theory (DFT) (Figure 5d) simulations have shown that a weak vdW force can exist between TMDC and graphene. Specifically, for MoS_2_, the binding energy is expected to be −23 meV per C atom regardless of the adsorption arrangement [48]. This weak vdW force leads to an enhanced electric dipole moment and dipole–dipole interaction at the TMDC/graphene interface which improves charge transfer at the interface and results in an enhanced SERS CM effect [21,49]. Although the mechanism needs further investigation, TMDCs nanostructures may allow LSPR via doping of photo-induced carriers and enhanced EM effect of SERS in a comparable way to the LSPR in nanocrystals and quantum dots [50,51]. That could explain the extraordinary sensitivity (R6G sensitivity of 2 × 10^−12^ M) (Figure 5e,f) of the MoS_2_ N-donut/graphene, which is higher than the best levels recently reported using the plasmonic metal nanostructure/graphene substrates [52].

### 3.3. SERS Enabled by Combined Metallic and Non-Metallic LSPR Nanostructures

TMDCs nanostructures may enable LSPR through the doping of photo-induced carrier in a similar way to the quantum dots and LSPR nanocrystals [50,51]. Therefore, TMDCs nanostructures can provide strong EM enhancement through implementation of non-metallic plasmonic nanostructures on 2D materials vdW heterostructures. This EM enhancement can be combined with the EM enhancement of metallic nanostructures to provide SERS with extra-high sensitivity. In recent work, Ghopry et al. explored Au nanoparticles (AuNPs) on TMDCs, specifically WS_2_-NDs, for the LSPR effect enabled by the enhanced dipole–dipole interaction at vdW interfaces in AuNPs/WS_2_-NDs/graphene heterostructures for SERS and reported strong EM which reveals the benefits of a superposition of LSPR effects from different nanostructures through the design of vdW heterostructures [53].

Three-dimensional planes of the electron localization function (ELF) of the stack of Au/WS_2_ heterostructure was investigated (Figure 6a) and revealed delocalized electrons in “C form” red contour configurations attaching S atoms to Mo atoms [54]. The superposition of the EM effect was confirmed by enhanced graphene Raman signatures (Figure 6b). The graphene G-peak intensity boosted by approximately 4.0- and 5.3-fold on AuNPs/graphene and on WS_2_-NDs/graphene, respectively, and boosted even further to 7.8-fold on the AuNPs/WS_2_-NDs/graphene over that of reference graphene sample. Figure 6c–f demonstrate the WS2 Raman map, AFM image, SEM image and EDS maps confirming the nanodisc shape and size, as well as the distribution uniformity of the AuNPs and WS2-NDs on top of graphene. This work showed enhancement factors of ~2.0 and 2.4 respectively based on the intensity of the R6G 613 cm^−1^ peak detected on the AuNPs/WS_2_-NDs/graphene with respect to that at the same intensity on WS_2_-NDs/graphene and AuNPs/graphene. This, therefore, illustrates the benefit of superposition to the SERS enhancements from the unique platform of non-metallic and metallic combined EM effect by plasmonic AuNPs and TMDC-NDs as shown in Figure 6g.

### 3.4. SERS Substrates with Both CM and EM Enhancement

The CM effect relies on the probe molecule/SERS substrate interface electronic structures. Heterostructures of 2D atomic materials would allow further tuning of the electronic structure of the SERS substrate for improved CM enhancement [45,46,47]. The EM and CM enhancements can be integrated on one SERS substrate implementing the plasmonic nanostructures on 2D atomic materials and their vdW heterostructures [31,39,55,56,57]. For instance, plasmonic AuNPs were fabricated on a large monolayer of graphene [24,39] using an in situ metal evaporation method [56]. This technique allows the manufacture of SERS substrates on a commercially compatible scale with extraordinary SERS sensitivity. Lu et al. made a SERS substrate of AuNPs/graphene, as shown in Figure 7a, where the AuNPs forms on graphene with dimension and morphology highly comparable to that on SiO_2_/Si substrates [39]. Using R6G probe molecules, they found a SERS enhancement 4 times greater than that on AuNPs alone (Figure 7b), confirming that the EM enhancement from AuNPs and the CM enhancement from graphene can be favorably combined. Employing a 633 nm laser, a sensitivity of 8 × 10^−7^ M has been achieved on R6G molecules (Figure 7c) [24].

Recently, a different stacking sequence of graphene and AuNPs was reported [52]. Xu et al. covered AuNPs with graphene (Figure 7d) and achieved a SERS sensitivity of 1.0 × 10^−11^ M for R6G on graphene/AuNPs (Figure 7d) employing the 532 nm resonant excitation. The explanation for the improved sensitivity is that the attachment area of the molecules on graphene would be higher in a graphene/AuNPs SERS substrate. It should be noted that the excitation laser wavelength could affect the SERS sensitivity. The sensitivity of the resonant SERS is typically many orders of magnitude higher than those situations without the resonance [41]. For example, the resonance wavelength of R6G is 532 nm. That suggests the stacking sequence may not be the only reason for the higher sensitivity in this work and may generate a minor difference in SERS. Using a similar idea, Chen et al. synthesized multilayer MoS_2_ onto Ag nanoparticles (AgNPs) (Figure 7e) via thermal decomposition and found SERS sensitivity up to 1 × 10^−9^ M of R6G on MoS_2_/AgNPs SERS substrate using 532 nm excitation (Figure 7f) [58]. On the other hand, Shorie et al. synthesized AuNP (using in situ deposition)/WS_2_ flakes (using liquid phase exfoliation) SERS substrates (Figure 7g) and reported sensitivity around 1 × 10^−8^ M using 532 nm excitation (Figure 7h) [59].

Table 1 summarizes R6G SERS substrates enhancement and sensitivity based on 2D atomic materials or their vdW heterostructures in comparison with metallic counterparts. It should be noted that the Raman excitation wavelength is an important parameter affecting the SERS sensitivity. Specifically, the resonant SERS would result in an increased Raman signal and consequently higher SERS sensitivity. For R6G dye molecules, the Raman excitation wavelength of 532 nm closely matches with the resonance absorption wavelength of the molecule. A higher SERS sensitivity with 532 excitation is expected compared to other non-resonant excitations, such as 633 nm. Moreover, Table 1 shows the use of plasmonic nanostructures enables SERS amplification via both EM and CM modes. The fact that non-metallic plasmonic nanostructures can offer equivalent or better SERS EM enhancement should be emphasized. For example, the best performance obtained on non-metallic MoS_2+_WS_2_ N-disc/graphene SERS substrate [44] was about a factor of 1.3 greater than that on metallic AuNPs/WS_2_ N-disc/graphene SERS substrate [53] and may be due to lower Ohmic loss on the LSPR TMD-NDs.

## 4. Applications of LSPR Nanostructures in Photodetection

Photodetectors are a crucial component in modern technology, with a wide range of applications including TV remote controls, fiber optic connections, video cameras, and astronomy. They are typically made of semiconducting materials, both inorganic and organic, and work by absorbing incident photons to generate electron–hole pairs and produce a photocurrent as a signal.

Recent advances in nanomaterials and technology have led to the development of high-performance nanohybrid photodetectors consisting of graphene and quantum nanostructures [40,60,61,62,63,64,65,66,67,68,69,70]. Theses nanostructures, which include quantum dots (0D) [61,71,72,73,74,75], nanotubes and nanowires (1D) [76,77], and nanosheets (2D) such as TMDCs [40,78,79], act as photosensitizers and generate an electric field across the interface with graphene, facilitating the transfer of photo-generated charge carriers. The implementation of graphene with its extraordinary charge carrier mobility, enhances the charge carrier transport in these nanohybrids, making them superior to counterparts based on nanostructures alone.

Various figures of merit are used to characterize the performance of a photodetector, including quantum efficiency, responsivity, noise, signal-to-noise ratio, specific detectivity, etc. [80]. Responsivity (R) is used to quantify the signal output (photocurrent, *I_ph_*) from the photodetector per unit optical power (*P_in_*) and is expressed as:(4)$$R=\frac{I_{ph}}{P_{in}}.$$

The photocurrent originates from the photo-generated carriers. In a photodiode, photocurrent is determined by the formula:(5)$$I_{ph}=\eta q\frac{P_{in}}{\hbar \omega }$$
where η is the quantum efficiency, ℏ is the Planck constant and ω is the light frequency. Quantum efficiency is η defined by the ratio between the number of electron–hole pairs generated and the number of photons injected. In other words, it represents the yield of charge carriers in the photodetector with a certain amount of photon incidents.

Achieving higher photoresponsivity is important to high performance photodetectors. This can be achieved through enhanced light absorption or external quantum efficiency in the photodetector using plasmonic nanostructures for light trapping. Metal nanoparticles (NPs) have been extensively adopted for light management in the visible spectrum [81,82,83]. On metal NPs, LSPR can be excited via incident light, and a strong evanescent EM field can be generated around the metal NPs within a short range of few tens to a few hundreds of nm [84,85]. The LSPR frequency is determined by free electron concentration in the metal, with minor tunability by the size and shape of the metal NPs as well as the external surrounding media [86]. An important result of the LSPR is the enhancement of the EM field surrounding metal NPs from several times to tens of times [86,87,88]. Therefore, this enhanced EM field can provide dramatically enhanced absorption in the photodetector, leading to an increased concentration of electron–hole pairs and thereby improved photoresponsivity.

### 4.1. Implementation of LSPR Nanostructures to Promote Exciton–Plasmon Coupling

Exciton–plasmon coupling refers to the interaction between excitons (bound electron–hole pairs) and plasmons (collective oscillations of conduction electrons) in a material system. This interaction occurs when the plasmon resonance of a nanoparticle or nanostructure overlaps with the absorption or emission spectrum of the active material, resulting in enhanced light–matter interaction and improved device performance. The coupling of excitons with plasmons can lead to energy transfer, charge transfer, or modification of the emission properties of the material system.

Directly embedding metal LSPR nanostructures to the photodetector is the easiest way to realize exciton–plasmon coupling. For example, Sun et al. reported enhanced responsivity by a factor of 1.96 on perovskite (400 nm thick)/graphene heterojunction photodetectors by embedding plasmonic AuNPs of ~40 nm in diameter directly underneath graphene [89] (Figure 8a). Wang et al. improved the photoresponsivity of a vertically structured perovskite photodetector by 59% by inserting Au nanorods in the perovskite layer [90] (Figure 8b).

However, merging metal NPs in the device with direct contact to sensitizer leads to several drawbacks that counteract the benefits of NPs plasmonic optical enhancement. Firstly, natural defect sites on the surface of metal NPs could trap both electrons and holes that are excited in the sensitizer layer. This leads to an increase in the recombination of photo-generated charge carriers [91]. Additionally, surface energy transfer can occur, in which photo-excited electrons recombine with holes in the sensitizer and transfer the excitation energy to neighboring metal NPs [92,93]. Both charge and energy transfers between the sensitizer and metal NPs can result in significant losses of photo-generated charge carriers, thereby weakening the plasmonic enhancement (Figure 8c). To address these issues, integrating plasmonic metal NPs into a photodetector must be deliberately designed to achieve optimal exciton–plasmon coupling and responsivity enhancement.

**Figure 8 micromachines-14-01393-f008:**
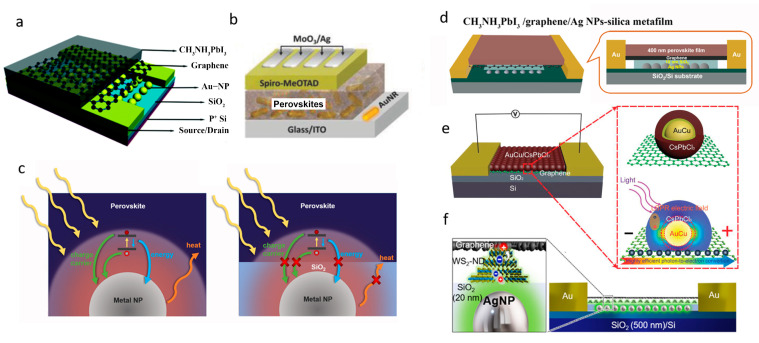
(**a**) Device structure of a CH_3_NH_3_PbI_3_/graphene/AuNPs hybrid photodetector. Reproduced with permission from Ref. [89]. Copyright 2009, Royal Society of Chemistry. (**b**) Device configuration of a CH_3_NH_3_PbI_3_/AuNPs hybrid photodetector. Reproduced with permission from Ref. [90]. Copyright 2018, Wiley. (**c**) Schematic diagrams of a metal NP embed in a perovskite thin film with a direct contact to perovskite, and a metal NP embedded in a SiO_2_ metafilm with a perovskite thin film atop. The yellow arrows represent incident light on the perovskite. The charge, energy and heat transfer effects are illustrated by green, blue and orange arrows, respectively, which can be effectively suppressed using the AgNPs embedded silica metafilm. (**d**) Schematic structure of a perovskite/graphene heterojunction photodetector on AgNPs-silica metafilm. Reproduced with permission from Ref. [94]. Copyright 2019, American Chemical Society. (**e**) Schematic image of AuCu/CsPbCl_3_ core/shell/graphene hybrid photodetectors with LSPR enhanced photoresponsivity. Reproduced with permission from Ref. [95]. Copyright 2020, Wiley. (**f**) Schematic illustration of the graphene/WS_2_-ND/AgNP metafilm heterostructure photodetector Reproduced with permission from Ref. [96]. Copyright 2019, American Chemical Society.

Liu et al. [94] reported a new device structure by implementing an AgNPs embedded SiO_2_ (AgNPs-silica) metafilm as the substrate of the perovskite/graphene photodetector, aiming to efficiently avail the plasmonic enhancement by suppressing the unfavourable charge, energy, and heat transfers. Specifically, an enhancement factor of 7.45 in the responsivity was obtained in perovskite/graphene photodetector on AgNPs-silica metafilm as compared to the counterpart without AgNPs integration (Figure 8d). A study by Alamri et al. [96] combined the plasmonic effects from TMDC-ND grown on graphene and AgNPs using an in vacuo process, producing a seven-fold enhanced photoresponse **(**Figure 8f).

In addition, metal/semiconductor core/shell structure design is another efficient way to improve the light absorption at a limited active shell thickness with low absorption capacity. A template modulated colloidal approach for synthesizing metal core (AuCu) and metal-halide perovskite shell (CsPbCl_3_) nanocrystals was reported as shown in Figure 8e [95]. Enhanced light absorption was observed in the CsPbCl_3_ shell with thickness of 2–4 nm, ascribed to the LSPR AuCu core with an average diameter of 7.1 nm. The LSPR AuCu core-induced light absorption of the perovskite shell enabled a remarkable 30-times-enhanced photoresponse in core/shell perovskite QDs/graphene nanohybrid photodetectors, compared with the counterparts with perovskite-only QDs.

### 4.2. LSPR Enabled by Metallic Nanostructures for Enhanced Photoresponse

Metallic LSPR nanostructures could be easily formed through deposition of a metal film and annealed in an inert gas or vacuum. This method leads to randomly distributed metal NPs, with diameters and LSPR wavelengths that can be tuned by controlling the film thickness. Complicated metal nanostructures could be generated via metal film deposition followed by electron beam lithography (EBL) [97]. In addition, some reports also demonstrated colloidal growth of Au nanorods, which can be mixed in the sensitizer [98,99].

Typically, noble metals Ag and Au are two promising plasmonic materials for the application in the photodetector, as the LSPR frequency of Ag and Au could be tuned over visible spectrum. Huang et al. fabricated a QD photodetector by linking an 80 nm thick Ag NP monolayer on a glass substrate with 3-aminopropyltriethoxylsilane [100]. They then drop-cast a CdSe/ZnS QD layer with a controllable thickness. By tuning the AgNP LSPR at λ = 400–600 nm, the device showed a 1.2–1.6-fold enhancement in responsivity for a 440 nm thick QD layer and a 2.4–3.3-fold enhancement for a 100 nm thick QD layer. Li et al. created a 2D plasmonic photodetector by encapsulating MoS_2_ on AuNPs [101]. The AuNPs were formed on a silicon substrate through annealing a gold film at 800 °C for 15 min, while the MoS_2_ was grown in a tube furnace at 650 °C with the substrate facing MoO_3_, resulting in the MoS_2_ being wrapped around the AuNPs. The resulting core/shell photodetector exhibited an enhanced electric field due to the LSPR at 600 nm, leading to a 10-fold enhancement of responsivity compared with planar MoS_2_ detectors.

Non-noble metal nanoparticles have garnered interest as an alternative to noble-metal nanoparticles due to their ability to induce LSPRs despite having large negative real and small imaginary dielectric functions. Lu et al. reported growing a ZnO nanorod array on a quartz substrate in a tube furnace followed by depositing aluminium nanoparticles via sputtering [102]. The aluminium nanoparticles had LSPRs at λ = 380 nm and increased the responsivity of the ZnO nanorod photodetector from 0.12 to 1.59 A/W under a 5 V bias. Another non-noble metal that has utilized plasmonic effects is copper. Wang et al. reported a graphene-CdSe nanoribbon heterojunction photodetector with ordered copper plasmonic nanoparticles added [103]. The copper nanoparticles were deposited onto the heterojunction using polystyrene spheres as a template and were later removed. The resulting device exhibited apparent LSPR enhancement in the red-light range of 700–900 nm.

### 4.3. LSPR of Doped-Semiconductor NCs

Colloidal semiconductor nanocrystals (NCs) with sizes on the order of a few tens of nanometres are also suitable to exploration of LSPR. The exhibition of LSPR in doped semiconductor NCs is due to their high concentration of free carriers that support the coherent collective charge oscillation under light excitation. As a result, their optical response can be explained by quasi-static dipole oscillators.

When the dimensions of colloidal semiconductor NCs are restricted to a few to tens of nanometres, the LSPR effect becomes much more pronounced due to the strong quantum confinement of the electrons within the NCs. This enhanced LSPR effect can be combined with the unique properties of quantum confinement in semiconductor NCs to provide a range of advantages for photonic and optoelectronic applications, including enhanced light absorption, increased photoluminescence, and improved quantum efficiency. Another major advantage of colloidal semiconductor NCs is their low fabrication cost, allowing for large-scale production of these materials. In addition, the composition, shape, size, as well as carrier doping of the NCs can be precisely controlled by adjusting the precursor solution used in their synthesis.

Controlling free-carrier concentration in semiconductor NCs through doping allows for the tuning of their LSPR frequency. Doping could be achieved through controlled heteroatom method, therefore inducing a targeted density of free carriers supporting collective oscillation on account of the discrepancy in the number of valance electrons between the enthetic atoms and the host atoms (i.e., the substituted atoms). For example, in 2018, Zandi et al. revealed that the resonant peak of the LSPR extinction spectrum of Sn:In_2_O_3_ (STO) NCs markedly shifted from ≈3000 to ≈5500 cm^−1^ with the nominal Sn doping level ranging from 1% to 10% [104].

Apart from heteroatom doping exerting polyvalent state combination, free carriers can also be intentionally introduced into semiconductors by finely tuning their atomic stoichiometric ratio. For example, the doping of Cu-based semiconductor NCs (Cu_2–x_S or Cu_2–x_Se) through variation the elemental ratio of Cu and S (or Se) has illustrated the LSPR effect-induced broadening of the absorption spectra from originally visible to near-infrared (NIR, 800–1000 nm) spectrum [51,105,106].

Various attempts have been made and reported in order to achieve better performance in photodetector through LSPR enhancement of Doped-Semiconductor NCs. Gong et al. reported a broadband photosensitizer by doping FeS_2_ nanocubes via induced growth defects or chemical composition variation, which not only result on enhanced absorption but also broadened spectral range due to red shift in the optical cutoff. The demonstrated strong LSPR effect has shown to enhance broadband light absorption covering the UV–Visible–SWIR (SWIR regards short-wave IR) [50,107]. This LSPR FeS_2_/graphene nanohybrid photodetector exhibits a high broadband photoresponse, with responsivities reaching 1.08 × 10^6^ A/W in the entire UV–Visible–SWIR range at a low operational voltage of 0.1 V. Similarly, various other doped semiconductor nanostructures, such as Cu_3−x_P [108], Cu_x_S [109] and MoO_x_ [110] have been developed into photodetectors by advancing the LSPR enhancement [111]. For example, Sarkar et al. obtained a MoSe_2_/GaAs heterojunction photodetector with improved performance through a defect-assisted formation of Cu_2−x_S nanostructures on MoSe_2_ [101], leading to an on/off ratio exceeding 1 × 10^5^ on the Cu_2_S@MoSe_2_/GaAs heterojunction photodetector. This is about two orders of magnitude higher than that of a pristine MoSe_2_/GaAs device (1 × 10^3^).

## 5. Summary and Future Perspectives

In summary, the exciting progress made in nanomaterials and atomic materials research in the last few decades has provided a promising platform for the design and fabrication of LSPR nanostructures. The key to the design of efficient LSPR nanostructures is associated with the excitation of plasmons on the surface of the nanostructures. This illustrates the importance of tuning the electronic structure and free carrier concentration in LSPR nanostructures. Among others, the 2D atomic materials and their vdW heterostructures are particularly interesting due to the unique electronic structures of these materials at the 2D limit and the tuning of the electronic structures via the selection of different atomic sheets for the stack, as well as engineering the corresponding vdW interfaces. As compared to the conventional metallic LSPR counterparts, the 2D LSPR nanostructures provide the advantages of a broader-band spectrum for LSPR, higher efficiency and lower cost. Using SERS substrates and photodetectors as two examples, excellent performance has been demonstrated via the adoption of 2D LSPR nanostructures, with some surpassing that of their metallic counterparts. Moreover, many methods in fabrication of 2D LSPR nanostructures are low cost and scalable for commercial applications. It should be noted that the research in the exploration of 2D LSPR nanostructures is only at the beginning, which can be accelerated through further understanding of the tunability of the vdW interface on the electronic structure and free carrier concentration in these 2D LSPR nanostructures. This understanding is crucial to the design of the 2D LSPR nanostructures and certainly will be an interesting topic of future research.

## Figures and Tables

**Figure 1 micromachines-14-01393-f001:**
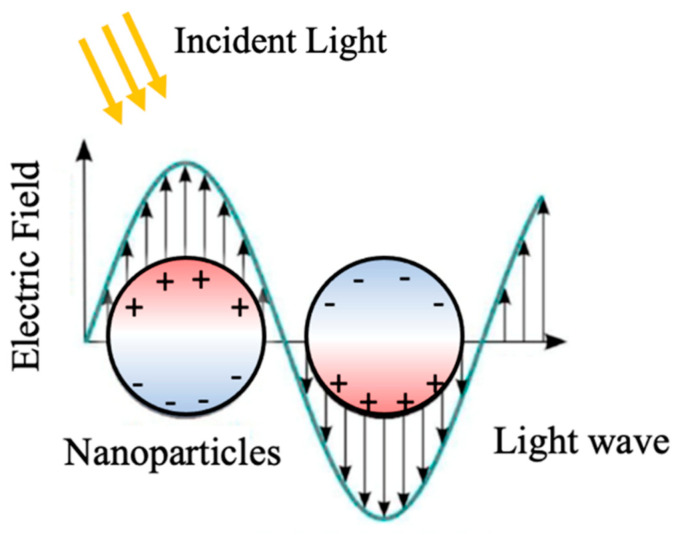
Schematic description of LSPR in metallic NPs through oscillation of free electrons on the NP surface.

**Figure 2 micromachines-14-01393-f002:**
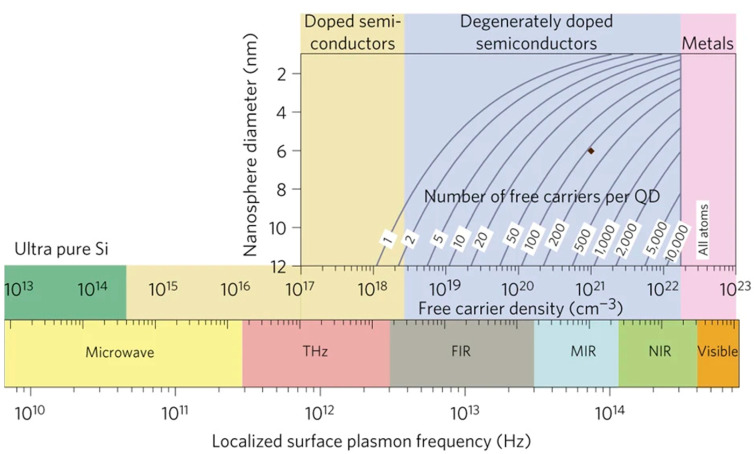
Localized surface plasmon resonance (LSPR) frequency dependence on free carrier density and doping constraints. The bottom panel shows the modulation of the LSPR frequency (*ω_sp_*) of a spherical nanoparticle by control of its free carrier concentration (*N*). LSPR frequency is estimated as: 1/2π $\sqrt{\left({Ne}^{2}/(\varepsilon_{o}m_{e}(\varepsilon_{\infty }+2\varepsilon_{m}\right)}$. The high frequency dielectric constant $\varepsilon_{\infty }$ is assumed to be 10, the medium dielectric constant $\varepsilon_{m}$ is set as 2.25 for toluene, and the effective mass of the free carrier $m_{e}$ is assumed to be that of a free electron. The e is the electronic charge and $\varepsilon_{o}$ is the permittivity of free space. The top panel shows a calculation of the number of dopant atoms required for nanoparticle sizes ranging from 2 to 12 nm to achieve a free carrier density between 10^17^ and 10^23^ cm^−3^. For LSPRs in the visible region, a material in which every atom contributes a free carrier to the nanoparticle, as for metals, is required. For LSPRs in the infra-red, carrier densities of 10^19^–10^22^ cm^−3^ are required. Below 10^19^ cm^−3^, the number of carriers (for a 10 nm nanocrystal) may be too low (<10) to support an LSPR mode. The brown diamond indicates the region of interest in the present study. Reprint with permission of [13], Copyright 2011, Springer Nature.

**Figure 4 micromachines-14-01393-f004:**
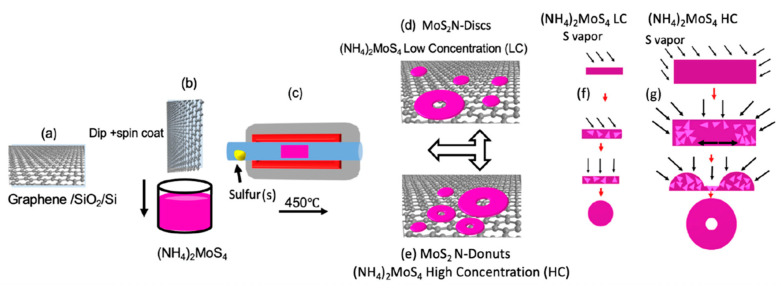
Schematic depiction of the synthesis process: (**a**) wet transfer of graphene on SiO_2_/Si substrates (**b**,**c**) MoS_2_ synthetization on graphene via the vapor transport process. (**d**,**e**) Synthesis of the MoS_2_ nanostructures: N-disc at low (**f**) and N-donut at high (**g**) (NH4)_2_MoS_4_ precursor concentration with the hypothetical growth mechanism where triangles denote MoS_2_ nuclei produced during the vapor transport annealing procedure. Reproduced with permission [43]. Copyright 2021, MDPI.

**Figure 5 micromachines-14-01393-f005:**
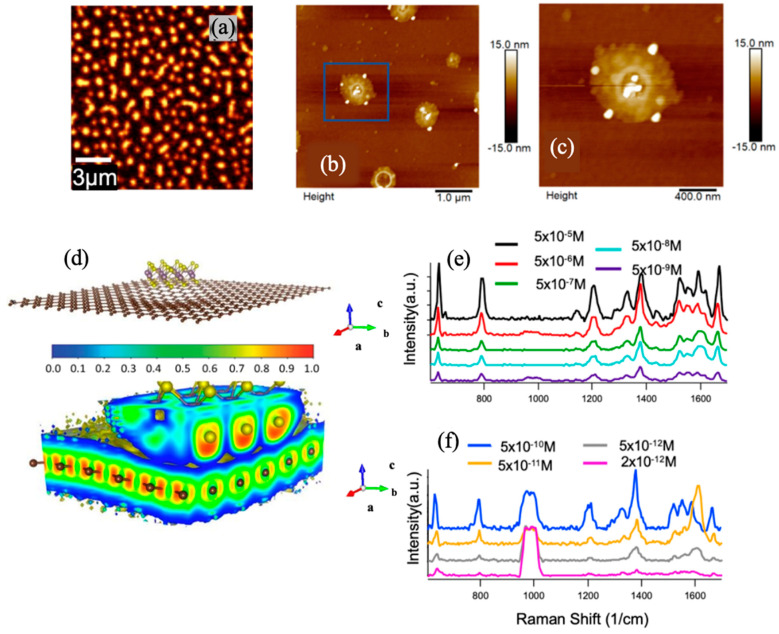
(**a**) Raman map of MoS_2_ (using A_1g_ mode). (**b**) Representative AFM images (5 μm×5 μm) of the sample with zoom in view (**c**). Reproduced with permission [43]. Copyright 2021, MDPI. (**d**) Vertical MoS_2_/graphene bilayer heterostructure (top) and the corresponding Electron Localization Function (ELF) (bottom) showing the localized electron concentration underneath the sulfur atom demonstrating the charge transfer occurrence. Reproduced with permission [41]. Copyright 2019, Wiley-VCH. (**e**) Raman spectra of the R6G molecules with different concentrations in the range of 5 × 10^−5^ M to 5 × 10^−9^ M (**c**), and (f) 5 × 10^−10^ M to 2 × 10^−12^ M on the MoS_2_ N-donut/graphene vdW heterostructures substrates. All spectra were collected using 532 nm excitation. Reproduced with permission [43]. Copyright 2021, MDPI.

**Figure 6 micromachines-14-01393-f006:**
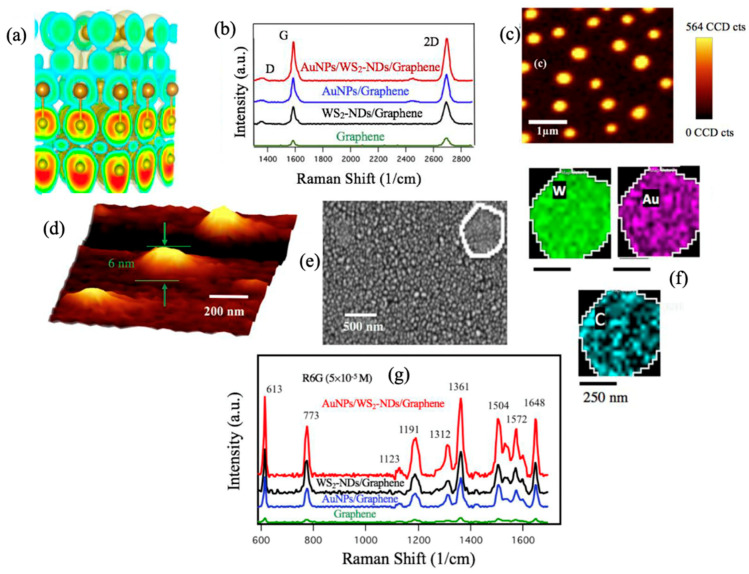
(**a**) 3D ELF plot of the stack of Au/WS_2_ heterostructure. Reproduced with permission [54]. Copyright 2019, ACS. (**b**) Graphene Raman spectra taken on four samples: graphene only, WS_2_NDs/graphene, AuNPs/graphene and AuNPs/WS_2_-NDs/graphene. (**c**) Raman map of WS_2_. (**d**) AFM image of the WS_2_-NDs; (**e**,**f**) an SEM image of an AuNPs/WS_2_-NDs/graphene sample and EDS maps of W, Au and C. (**g**) Raman spectra of R6G molecules with a concentration of 5 × 10^−5^ M on different substrates. Reproduced with permission [53]. Copyright 2020, ACS.

**Figure 7 micromachines-14-01393-f007:**
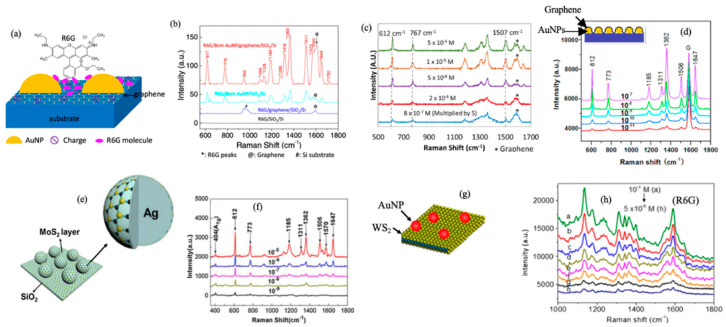
(**a**) Schematic description of AuNP/graphene SERS substrate with R6G probe molecules attached. (**b**) Comparison of Raman spectra of R6G probe molecules attached on four different substrates: AuNPs/graphene/SiO_2_/Si, AuNPs/SiO_2_/Si, graphene/SiO_2_/Si and bare SiO_2_/Si, respectively. Reproduced with permission [39]. Copyright 2015, Elsevier. (**c**) Raman spectra of R6G molecules at different concentrations in the range of 5 × 10^−5^ to 8 × 10^−7^ on AuNP/graphene SERS substrate. Reproduced with permission [24]. Copyright 2017, Elsevier. (**d**) Schematic description of graphene/AuNPs SERS substrates (top-left inset), and the Raman spectra of R6G molecules on the graphene/AuNP SERS substrates of concentrations in the range of 10^−7^ to 10^−11^. Reproduced with permission [52]. Copyright 2016, Elsevier. (**e**) Schematic diagram of MoS_2_/AgNPs hybrid system (**f**) Raman spectra of R6G molecules of different concentrations from 5 × 10^−5^ to 8 × 10^−9^_._ Reproduced with permission [58]. Copyright 2016, Elsevier. (**g**) Schematic showing the Au-WS_2_ nanohybrid SERS platform, (**h**) Raman spectra of R6G molecules using concentrations from 5 × 10^−4^ to 8 × 10^−8^ on Au-WS_2_ nanohybrid SERS. Reproduced with permission [59]. Copyright 2018, Microchimica Acta.

**Table 1 micromachines-14-01393-t001:** Performance comparison of SERS substrates based on 2D materials and vdW heterostructures and their hybrids with metallic LSPR nanostructures using R6G as the probe molecules.

Substrates Materials	SERS Substrates Design	Enhancement Factor	Sensitivity	Excitation Wavelength	Ref.
AuNPs/graphene	Metallic LSPR	-	8 × 10^−7^ M	633 nm	[24]
graphene/AuNPs	Metallic LSPR	4.8 × 10^7^ (EM + CM)	10^−11^ M	532 nm	[52]
MoS_2_/AgNPs	Metallic LSPR	3.75 × 10^4^ (EM + CM)	10^−9^ M	532 nm	[58]
AuNP/WS_2_	Metallic LSPR	6.78 × 10^6^ (EM + CM)	1 × 10^−8^ M	532 nm	[59]
AuNPs/graphene	Metallic LSPR	4, compare to SERS on AuNP (EM + CM)	-	633 nm	[39]
AuNPs/graphene	Metallic LSPR	-	8 × 10^−7^ M	633 nm	[24]
AuNPs/MoS_2_ (continuous)/graphene	Metallic LSPR	-	5 × 10^−10^ M 5 × 10^−8^ M	532 nm 633 nm	[54]
WS_2_ N-disc/graphene	Non-metallic LSPR	~8, compared to SERS on WS_2_ continuous layer and a graphene single layer (EM + CM)	5 × 10^−11^ M	532 nm	[41]
MoS_2_ N-disc/graphene	Non-metallic LSPR	~9, compared to SERS on MoS_2_ continuous layer and a graphene single layer (EM + CM)	5 × 10^−12^ M	532 nm	[41]
MoS_2+_WS_2_ N-disc/graphene	Non-metallic LSPR	14–17, compared to SERS on graphene single layer (EM + CM)	7 × 10^−13^ M	532 nm	[44]
AuNP/WS_2_ N-disc/graphene	Metallic and non-metallic LSPR	12.2, compared to on graphene	10^−12^ M	532 nm	[53]
MoS_2_ N-donut/graphene	Non-metallic LSPR	~20 compared to on graphene	2 × 10^−12^ M	532 nm	[43]

## Data Availability

Not applicable.

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
