# Peer review of "Metallic and Non-Metallic Plasmonic Nanostructures for LSPR Sensors"

_micromachines, 2023, doi:10.3390/mi14071393_

Round 1

Reviewer 1 Report

Manuscript No:  micromachines-2418571

Title:  Metallic and Non-Metallic Plasmonic Nanostructures for LSPR Sensors

Authors:  Judy Z. Wu1, Samar Ali Ghopry, Bo Liu, and Andrew Shultz

A. Overview

1. In this manuscript the authors present an extended and comprehensive review on plasmonic nanostructures for localized surface plasmonic resonance.

2. Abstract – “Surface enhanced plasmonic resonance (LSPR)” ?? – authors must read systematically and carefully the manuscript.

3. The contents are expressed clearly; the manuscript is well organized, and it is written in reasonable English.

Nevertheless, reading of the manuscript is needed as several typos and grammar issues ca be found in the manuscript.

3. The authors have acknowledged research on this topic, however most of the references are older than 5 years. Please provide a few newer references (5 years at most)

4. Quality of all figures must be improved (can’t see nothing in figure 2 … )

B. Overall assessment

The review presented here is very interesting and has potential for help authors and research groups. In my opinion the manuscript can be published after corrections.

C. Review Criteria

1. Scope of Journal

Rating: Medium

2. Novelty and Impact

Rating: Medium

3. Technical Content

Rating: Medium

4. Presentation Quality

Rating: Medium

The contents are expressed clearly; the manuscript is well organized, and it is written in reasonable English.

Nevertheless, reading of the manuscript is needed as several typos and grammar issues ca be found in the manuscript.

Author Response

Response to Reviewer 1

  1. In this manuscript the authors present an extended and comprehensive review on plasmonic nanostructures for localized surface plasmonic resonance.

Response: We thank our reviewer for spending valuable time in reviewing this manuscript and for his/her valuable comments according to which the manuscript has been revised. 

  1. Abstract – “Surface enhanced plasmonic resonance (LSPR)” ?? – authors must read systematically and carefully the manuscript.

Response: We sincerely apologize for this careless error. The correct definition of LSPR has been  implemented in the abstract of the revised manuscript. 

  1. The contents are expressed clearly; the manuscript is well organized, and it is written in reasonable English.

Nevertheless, reading of the manuscript is needed as several typos and grammar issues ca be found in the manuscript.

Response: We thank our reviewer for the comment and have proofread the manuscript for errors and inconsistencies. We sincerely hope the revised manuscript meets the publication standard for Micromachines.   

  1. The authors have acknowledged research on this topic, however most of the references are older than 5 years. Please provide a few newer references (5 years at most)

Response: We agree with our reviewer that the references should be up to date. In the revised manuscript, 10 references in 2021-2023 have been added.   

  1. Quality of all figures must be improved (can’t see nothing in figure 2 … )

 Response:  The quality of figures in the original version of the manuscript may have been compromised during the conversion to the PDF format. To address this, we have enhanced the resolution of all figures, including Figure 2, to 600 dpi, surpassing the minimum requirement of 300 dpi by the journal of Micromachines.

5. The review presented here is very interesting and has potential for help authors and research groups. In my opinion the manuscript can be published after corrections.

Response: We thank our reviewer for the positive and constructive comments. We have made revisions accordingly to improve the quality of this manuscript. 

Reviewer 2 Report

The authors reviewed and discussed the recent development of SPR, LSPR and SERS technology based on semiconductor materials, especially for the 2D atomic materials (graphene) and heterostructures. Non-metallic LSPR nanostructures have been widely concerned for their free-dependence for local heating effect. I recommend revision for this submission for its possible publication in Micromachines. Some comments can be found below:

1.       Gramma and typos problems should be checked and fixed, for example: line 34 “can have increased”; line 118 “can be understand using a simple” etc.;

2.       Equations numbers should be unified and annotated in order;

3.       Please check and fix the sub-section numbers in Section 4;

4.       The authors said the introduction of non-metallic materials have expanded the spectra range of LSPR. Some examples should be given and compared;

5.       It is noticed that the references in recent 5 years are limited (~10 more maybe), more works are suggested to include.

Author Response

Response to reviewer 2

The authors reviewed and discussed the recent development of SPR, LSPR and SERS technology based on semiconductor materials, especially for the 2D atomic materials (graphene) and heterostructures. Non-metallic LSPR nanostructures have been widely concerned for their free-dependence for local heating effect. I recommend revision for this submission for its possible publication in Micromachines. Some comments can be found below:

  1. Gramma and typos problems should be checked and fixed, for example: line 34 “can have increased”; line 118 “can be understand using a simple” etc.;

Response: We thank our reviewer for the comment and have proofread the manuscript carefully for errors and inconsistencies. Specifically, the English errors on line 34 and line 118 have been fixed. We sincerely hope the revised manuscript meets the publication standard for Micromachines.  

  1. Equations numbers should be unified and annotated in order;

Response: We sincerely apologize for the inconsistencies that have been fixed in the revised manuscript.  

  1. Please check and fix the sub-section numbers in Section 4;

Response: Done.  

  1. The authors said the introduction of non-metallic materials have expanded the spectra range of LSPR. Some examples should be given and compared;

Response: A few examples have been discussed on Page 14 (line 492-495 and line 497-502) with the corresponding references.  

  1. It is noticed that the references in recent 5 years are limited (~10 more maybe), more works are suggested to include.

Response: We agree with our reviewer that the references should be up to date. In the revised manuscript, 10 references in 2021-2023 have been added.   

Round 2

Reviewer 2 Report

The authors have replied all comments and improved the manuscript.